# Communication to adult patients undergoing cancer care by non-specialist nurses: a scoping review protocol

Chiyembekezo Kachimanga  , Jennifer McGlashan, Nicola Cunningham, Louise Hoyle

University of Stirling, Stirling, UK

**Correspondence to**
Dr Chiyembekezo Kachimanga;
chk00095@students.stir.ac.uk

## ABSTRACT

**Introduction** Little is known regarding how non-specialist nurses communicate with patients living with cancer when the patients are receiving care outside of their cancer units/teams. This scoping review aims to identify, examine and report on the currently available evidence about communication by non-specialist nurses when caring for adults living with cancer outside of their cancer care unit/teams.

**Methods and analysis** A scoping review following the JBI methodology for scoping reviews will be conducted. We will search for empirical studies that meet the inclusion criteria in six databases (MEDLINE, PubMed, CINAHL, Embase, Scopus and PsycINFO). Handsearching in references of included articles will be performed to find additional articles. The population of interest will be non-specialist nurses. Three concepts will be explored, namely (1) all adult patients living with cancer, (2) a focus on three stages of the cancer continuum of care (cancer diagnosis, treatment and survivorship) and (3) a focus on communication between non-specialist nurses and patients living with cancer. We will include studies describing all healthcare settings outside patients' specialised cancer units or oncology teams. After article selection, two reviewers will independently screen titles and abstracts and perform a full-text article review, risk of bias assessments and data extraction. A third reviewer will resolve all disagreements. A narrative summary will provide an overview of how the results relate to the research aims and questions. The included articles will be limited to English and published between 2012 and 2023.

**Ethics and dissemination** No ethical approval is required since we will use publicly available empirical research sources. This review will provide current research on communication by non-specialist nurses with patients with a cancer diagnosis outside of an oncology setting, evidence that will support effective communication. As such, we aim to disseminate the findings in academic conferences and peer-reviewed journals.

## STRENGTHS AND LIMITATIONS OF THIS STUDY

⇒ This will be the first review that provides a comprehensive summary of communication by non-specialist nurses with patients with a cancer diagnosis outside of an oncology setting.
⇒ The scoping review will be systematic and transparent as it will be guided by JBI methodology for scoping reviews and will be reported using the checklist for Preferred Reporting Items for Systematic Reviews and Meta-Analyses extension for Scoping Reviews.
⇒ The review will only focus on empirical studies published in academic journals and will exclude articles published in grey literature.
⇒ The review will be limited to studies published in English between 2012 and 2023.

## INTRODUCTION

Cancer is one of the most commonly diagnosed non-communicable diseases and the leading cause of death worldwide.[1] In 2020 alone, over 20 million people were diagnosed with cancer, and nearly 17% of all deaths were caused by cancer.[2] Cancers affect all regions of the world, and the incidence of cancers will increase by up to 50% by 2040.[3]

The higher morbidity and mortality of cancer require appropriate care at all health system levels. Cancer care is provided by a multidisciplinary team from the community to tertiary care facilities. Among the essential healthcare providers are the nurses, the most numerous healthcare providers working with people diagnosed with cancer.[4] Whether generalists or specialists, nurses play essential roles in managing patients diagnosed with cancer through screening, treatment, survivorship or end-of-life care.[5] Nurse-led cancer interventions, which include direct patient care, education, counselling and care coordination, improve high-quality patient-centred cancer care.[6] Patients with cancer diagnoses need care from a well-trained, updated and skilled nursing workforce.[7]

Patients living with cancer require effective communication from their healthcare providers. Effective communication means providing all necessary information to allow

a good understanding of their cancer diagnosis and managing emotional responses stemming from an awareness of the diagnosis.[8] Effective communication also helps patients with a cancer diagnosis make informed care decisions, fosters strong relationships between patients and their providers,[9] improves clinical outcomes and improves patients' care experience.[10] Furthermore, effective communication enhances patient satisfaction and quality of life, reduces providers' stress and burn-out, improves adherence to cancer care, and eases caretaker burden.[9 11 12]

Communication with patients by healthcare providers is bidirectional and provided throughout the cancer care continuum.[13] The bidirectional nature entails that healthcare providers provide information that allows patients to make decisions based on their understanding of the disease, and patients are empowered to make decisions that will ultimately improve their care.[8] The cancer continuum depicts the steps involved in cancer care and control and includes (1) prevention and risk reduction, (2) screening, (3) cancer diagnosis, (4) treatment, (5) survivorship and (6) end-of-life care.[14] Throughout this cancer continuum, communication is provided by all healthcare providers involved in patient care. Communication with patients with a cancer diagnosis does not only happen when patients receive care from their oncology teams but also during any encounter with healthcare providers, including care outside their oncology teams, emergency settings and non-cancer related clinical visits.[15 16] Regardless of the setting of clinical encounters, communication should be continuous and reinforced over time.

Effective communication is widely recognised as one of the dimensions of quality in cancer care.[17] However, despite this recognition, effective communication remains one of the most common unmet needs for patients with cancer.[9] Effective communication goes beyond providing understandable information; it involves communicating in a patient-centred approach—a process where communication is provided sensitively, with respect to patient's autonomy and involving patients in the decision-making process.[18 19]

Communication in cancer care is a learnt clinical skill and goes beyond medical interviews due to the complexity and life-threatening issues commonly discussed with patients living with cancer.[20] As such, for oncologists and other healthcare providers working in cancer units, rigorous communication training should be provided as part of their initial and routine training.[21 22] However, even if communication training is in place, evidence suggests that trained healthcare providers need help with effective communication. For example, a study in cancer care settings in Africa found that only 40% of nurses and 20% of physicians had formal communications training.[23] Similar studies in Kenya and Belgium have also shown that lack of communication training is a major challenge in cancer care.[24–26] In these studies, emphasis was placed on expanding and improving communication training

during pre-service training, providing further guidance and mentorship during cancer care coordination and adapting communication to different contexts and/or cultural backgrounds.

Patients with a cancer diagnosis may require care outside their usual care teams, and nurses play a more prominent role in this context. Nurses, including non-specialised nurses, are usually the first-line workers, they spend most of their time with clients and interface with patients with cancer during outpatient and in-patient services.[27 28] Non-specialist nurses also have essential roles in cancer care which include direct nursing care, patient navigation, educating other healthcare providers and care coordination.[29]

To ensure good outcomes, non-specialist nurses must be equipped with practical communication skills when caring for an individual diagnosed with cancer. Little is known regarding how non-specialist nurses communicate with patients living with cancer when receiving care outside their cancer units/teams. This scoping review aims to understand non-specialist nurses' roles and skills in effectively communicating with adult patients living with cancer.

### Aims

The main objective of this scoping review will be to identify the roles and skills of non-specialist nurses in providing effective communication to adult patients with a diagnosis of cancer receiving any health services outside of their cancer care units. An initial scoping search was conducted in four databases (PROSPERO, PubMed, JBI systematic review register and Cochrane database) to check if any review is in progress or has been published on this scoping review objective. No review similar to this topic is being conducted or has been published (online supplemental appendix 1). This scoping review will answer four main questions:

1. What are the roles of non-specialist nurses in providing effective communication to adult patients with a diagnosis of cancer attending any health service outside of their cancer units?
2. When do non-specialist nurses communicate with adult patients with cancer attending any health services outside of cancer units?
3. How do non-specialist nurses communicate with adult patients with cancer attending any health services outside of cancer units?
4. What communication skills do they have to effectively communicate to adult patients with cancer attending any health services outside of cancer units?

### METHODS AND ANALYSIS

To identify, examine and report on the currently available evidence about communication by non-specialist nurses when caring for an individual with a diagnosis of cancer, the most appropriate approach to gathering and synthesising evidence will be a scoping review. This review will be

**Table 1** Inclusion and exclusion criteria

| Inclusion criteria | Exclusion criteria |
|---|---|
| 1. Studies reporting on non-specialist nurses. Non-specialist nurses are not specifically trained to provide expert advice and care for patients with cancer and do not have formal certification in cancer care.<br>2. Patients living with cancer aged 18 years and above.<br>3. Patients living with cancer in any of the three stages of the cancer continuum of care (diagnosis, treatment and survivorship)<br>4. Studies reporting on communication between non-specialist nurses and patients diagnosed with cancer. We will explore the following concepts:<br>a) Experiences and roles of non-specialist nurses in communicating with patients with cancer<br>b) Context where communication between non-specialist nurses and patients with cancer occurs<br>c) How communication (verbal and non-verbal) occurs between non-specialist nurses and patients with cancer<br>d) Training and skills in communicating with patients with cancer by non-specialist nurses<br>5. Communication occurs when patients with cancer access services outside of the cancer unit/care teams<br>6. Studies published in the English language<br>7. Studies conducted between 2012 and 2023<br>8. Qualitative, quantitative or mixed-methods study designs | 1. Studies reporting on specialist nurses, student nurses or other healthcare workers.<br>2. Studies reporting on cancer screening, end-of-life care or palliative care stage of cancer continuum of care<br>3. Studies not reporting on communication.<br>4. Studies reporting on patients that cannot communicate or have limited capacity to communicate, for example, dementia and Alzheimer's disease. |

guided by the Joanna Briggs Institute (JBI) methodology for scoping reviews.[30] The JBI methodology for scoping reviews was advanced and refined from two previous scoping review guidance from Arksey and O'Malley[31] and Levac *et al*.[32] Furthermore, we will use the checklist for Preferred Reporting Items for Systematic Reviews and Meta-Analyses extension for Scoping Reviews (PRISMA-ScR).[33] We registered the protocol in Open Science Framework registry (https://osf.io/3pmt9).

### Inclusion criteria

To define the scope and inclusion criteria of the review, we have used the population, concepts and context (PCC) framework.[30 34] The population of interest will be non-specialist nurses. Three concepts will be explored, namely (1) all adult patients (18 years and above) living with cancer, (2) a focus on three stages of the cancer continuum of care (cancer diagnosis, treatment and survivorship) and (3) a focus on communication between non-specialist nurses and patients living with cancer. In context, we will include studies conducted in any geographical location describing all healthcare settings outside patients' specialised cancer units or oncology teams. Table 1 details the inclusion and exclusion criteria.

### Sources of evidence

The review will consider empirical studies from peer-reviewed articles that used quantitative, qualitative and mixed-method designs. All designs within these types of studies will be included. All other study types, including review studies, commentaries, policy-related documents and conference proceedings, will be excluded. We will also handsearch references of included articles to identify additional articles. As the authors are proficient in the English language only, we will include studies published in the English language only. We also plan to review recent articles on communication; hence, we will restrict studies published in the last decade (between January 2012 and December 2023).

### Search strategy and information sources

This review aims to identify published empirical studies. The search strategy will follow a three-step process as recommended by the JBI methodology for scoping reviews.[35]

In step 1, an initial limited search was undertaken in October 2022. The following concepts were used as the initial starting point: "non-specialist nurse" AND Cancer AND communication. Each concept term was expanded by looking at synonyms and identifying common terms used in the literature.[21 36 37] Using the EBSCOhost platform for Medline, CINAHL and PsycINFO databases, appropriate medical subject headings and free text for each concept were developed.[38] The 'expand' option to expand some of the subject headings was used as appropriate. CK designed the initial search terms, which LH validated. Table 2 provides an overview of the sample search terms used at this initial point, and online supplemental appendix 2 provides the results of this limited search. The titles, abstract and index terms will be analysed from this initial search to inform the final search strategy.

| Table 2 | Sample search terms |
|---|---|
| 1 | Describing the population of this study |
| Participants | (MH "Nursing Staff+") OR (MH "Nurses+") OR (MH "Nursing+") OR (MH "Nursing Care+") OR (MH "Nurse Practitioners+") OR (MH "Nursing Staff, Hospital+") OR "nursing staff" OR nurs* OR "nursing care" OR "nurse practitioner*" OR "hospital nurse*" OR "non-speciali#ed nurse*" |
| 2 | Describing the concepts –Cancers |
| Cancers | (MH "Neoplasms+") OR neoplasm* OR neoplasia Or tum#r* OR "Malignant neoplasm*" OR Malignan* OR cancer* OR "cancer patient*" |
| 3 | Describing the concepts-communication and experiences |
| Communication and communication experience | (MH "Communication+") OR (MH "Communication Methods, Total") OR (MH "Disclosure+") OR communication OR "information exchange" OR "communication pattern*" OR "communication strategy*" OR "communication method*" OR "communication technique*" OR disclosure OR interaction* OR experience* |
| 4 | Combining all concepts |
| | 1 AND 2 AND 3 |
| 5 | Limiter 1: Date between 2012 and June 2022 |
| 6 | Limiter 2: adult 18 years and above |
| 7 | Limiter 3: English publication |

In step 2, we will perform another search using the developed search strategy from the previous step. A comprehensive search in the MEDLINE, PubMed, CINAHL, Embase, Scopus and PsycINFO databases will be performed. In step 3, additional studies will be supplemented by handsearching the references of included sources for additional citations.

### Study selection

After a comprehensive data search in the selected databases, we will export all articles to EndNote software V.20, where all duplicates will be removed.[39] Afterwards, all articles will be exported to Covidence software for article screening, full-text review and data extraction.[40] Two reviewers will independently review the title and abstracts to identify articles eligible for full-text review, conduct a full-text review to identify articles eligible for data extraction and perform data extraction based on a piloted data extraction tool. A third reviewer will resolve all disagreements during the title and abstract screening, full-text screening and data extraction. The full text of retrieved sources that do not meet the inclusion criteria will be excluded, but a note will be made for the reason for exclusion. The searching, screening and selection will be documented using a PRISMA-ScR flow diagram (figure 1). The flow diagram will be updated once the review has been completed.

### Data extraction and assessment of methodological quality

We will document the data based on a standardised data collection tool. Before data collection, the tool will be piloted on a selection of articles eligible for the title and abstract screening.[30] Two reviewers will independently perform the extraction, followed by a discussion to resolve differences and create the final data collection tool (see

table 3 for sample data collection tool). We plan to extract the following data based on the eligibility criteria:
1. Details of the article: author, year and study design.
2. Context: country of the study and healthcare setting.
3. Cancer care continuum.
4. Types and roles played by non-specialist nurses in cancer care.
5. Timing, frequency and contexts of cancer communication.
6. Communication methods used.
7. Communication training and ongoing skills (preservice and in-service skills) acquired by non-specialist nurses.

Despite prespecifying the type of data we will collect, we will adjust the data that will be collected during data extraction if necessary. All modifications to the data collection will be described during the final report of the scoping review.

Although it is not mandatory to perform risk of bias assessment in scoping reviews, we will include risk of bias assessment in this scoping review.[41 42] Risk of bias assessment is an appraisal of the methodological quality of the included studies in order to inform the synthesis of the collected data.[43] In this review, we will use risk of bias assessment to identify studies that should be included with the purpose of including studies with low risk of biases. Appropriate tools will be identified using JBI critical appraisal tools (https://jbi.global/critical-appraisal-tools). Two reviewers will independently perform a risk of bias assessment, and a third reviewer will resolve all disagreements.

### Presentation of results and expert consultation

We plan to map the evidence based on PCC using both quantitative and qualitative study designs. For quantitative

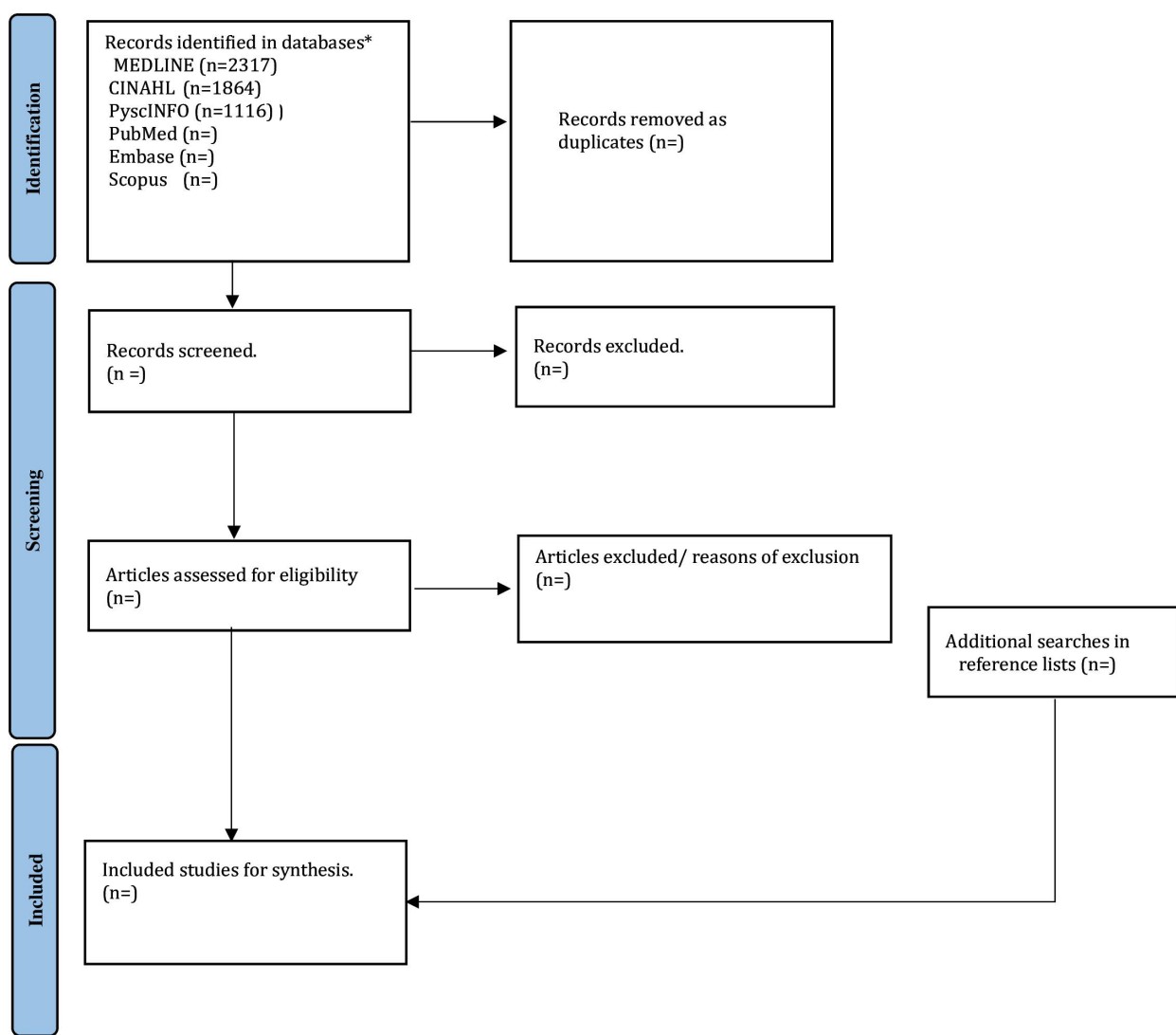

**Figure 1** PRISMA flow diagram. PRISMA, Preferred Reporting Items for Systematic Reviews and Meta-Analyses. *The search results will be updated by the end of the review.

designs, a narrative summary will be written to provide an overview of how the results relate to the research aims and questions. For qualitative designs, descriptive content analysis will be used. Tables and figures will be used to provide an overview of extracted data relevant to the research questions. We plan to continue the comprehensive literature search in April 2024 and finalise the review by March 2025.

During the comprehensive literature search and data extraction, we will seek additional support from subject-specific experts. We will seek an expert information specialist to validate the final search strategy. We will also seek support from a cancer expert to validate the findings prior to final dissemination.

### Patient and public involvement
There is no patient or public involvement in this study.

### DISCUSSION
To our knowledge, this will be the first review that looks at communication by non-specialist nurses with patients with a cancer diagnosis outside of an oncology setting. Patients living with cancer require comprehensive information

| | | | | | | | | |
|---|---|---|---|---|---|---|---|---|
| **Table 3** Sample data collection tool | | | | | | | | |
| **Author, year** | **Study design** | **Participants** | **Context continuum of care** | **Cancer / roles of non-specialist nurses** | **Type /roles of non-specialist nurses** | **Timing of cancer communication** | **Communication methods** | **Communication training/skills** |
| – | – | – | – | – | – | – | – | – |

regarding their diagnosis, which is often conveyed during provider–patient communication. While effective communication brings hope, peace of mind and trust in healthcare providers, ineffective communication leads to suboptimal cancer care, increased patient and caregiver distress and lack of trust in healthcare providers.[20]

Understanding the communication between non-specialised nurses and patients with a cancer diagnosis has implications for practice, policy and research. From a practical point of view, this review will highlight gaps in cancer communication, which may inform areas where clinical guidelines and training need to be included. The results may also influence policy-makers by introducing narrative evidence on best practices or gaps in cancer communication by non-specialised nurses. Finally, the study will help identify gaps in the literature on cancer communication. Based on these gaps, further studies, including systematic reviews, may be developed to understand specific aspects of cancer communication.

## ETHICS AND DISSEMINATION

Ethical approval will not be needed as we will use publicly available published empirical studies. We plan to disseminate the results in nursing-specific conferences and other academic conferences. The final report will be published in peer-reviewed academic journal.

**Contributors** CK conceptualised the study, developed the methods, developed the preliminary search strategy, wrote the first draft, critically reviewed the draft manuscript and approved the final draft for submission. LH conceptualised the study, developed the methods, developed the preliminary search strategy, critically reviewed the draft manuscript and approved the final draft for submission. JM conceptualised the study, critically reviewed the draft manuscript and approved the final draft for submission. NC provided critical inputs to methods and editing of this protocol, critically reviewed the draft manuscript and approved the final draft for submission.

**Funding** The authors have not declared a specific grant for this research from any funding agency in the public, commercial or not-for-profit sectors.

**Competing interests** None declared.

**Patient and public involvement** Patients and/or the public were not involved in the design, or conduct, or reporting, or dissemination plans of this research.

**Patient consent for publication** Not applicable.

**Provenance and peer review** Not commissioned; externally peer reviewed.

**ORCID iD**
Chiyembekezo Kachimanga http://orcid.org/0000-0003-3507-9591

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
