## [Reviewer comments · BMJ Open]

ARTICLE DETAILS

TITLE (PROVISIONAL)	Communication to Adult Patients Undergoing Cancer Care by Non-Specialist Nurses: a Scoping Review Protocol
AUTHORS	Kachimanga, Chiyembekezo; McGlashan, Jennifer; Cunningham, Nicola; Hoyle, Louise

VERSION 1 – REVIEW

REVIEWER	Xu, Zhijie Zhejiang University School of Medicine Second Affiliated Hospital, Department of General Practice
REVIEW RETURNED	25-Dec-2023

GENERAL COMMENTS	This scoping review is to identify the roles and skills of non-specialist nurses in providing effective communication to adult patients with a diagnosis of cancer receiving health services. This scoping review is to identify the roles and skills of non-specialist nurses in providing effective communication to adult patients with a diagnosis of cancer receiving health services. The authors proposed clear and important research questions of the scoping review, and the search strategy and inclusion/exclusion criteria is comprehensive and accurate. Some recommendations could be considered to further improve the expression of the protocol. A) It is advisable for the authors to incorporate the Preferred Reporting Items for Systematic Reviews and Meta-Analyses (PRISMA) flow diagram within the protocol to transparently convey the search outcomes and document the rationale behind article exclusion. Furthermore, it could be stated that an updated flow diagram will be provided upon the scoping review's completion. B) I recommend that the authors consult the methodological guidance provided by Arksey and O'Malley, which delineates a six-stage framework encompassing: (1) identifying the research question; (2) identifying relevant studies; (3) selecting studies; (4) charting the data; (5) collating, summarising and reporting results; and (6) consulting with stakeholders. Additionally, referencing Levac et al.'s recommendations could serve to refine and enhance the application of this framework. Arksey H, O'Malley L. Scoping studies: towards a methodological framework. Int J Soc Res Methodol 2005;8:19–32. Levac D, Colquhoun H, O'Brien KK. Scoping studies: advancing the methodology. Implement Sci 2010;5:69. C) (Page 13, line 44) The presentation of results could be expanded to include critical specifics. For instance, the methodologies for data analysis should be elaborated upon, including the tools intended for
--

	content analysis, particularly as the review includes both qualitative and quantitative research necessitating a robust mixed-methods analysis. D)Engaging stakeholders in consultation may yield further profound insights into the literature review. While this step is not mandatory, disseminating the findings amongst stakeholders could substantially aid in the interpretation and practical application of the results.
--	---

REVIEWER	Majamanda, Maureen Kamuzu University of Health Sciences, Child Health Nursing
REVIEW RETURNED	29-Dec-2023

GENERAL COMMENTS	This proposed scoping review addresses a very important area especially in contexts where cancer care is mostly provided by general nurses who are non-specialists. The protocol can be strengthened by addressing the proposed comments. Some references are older than 10 years. where possible get current references. the old ones can only be maintained if they have some relevant information which cannot be accessed in any recently published articles. Include funding details. Consider registering this protocol on Open Science Framework. Comment to the author  1. Lines 6-15 in abstract, the part of the sentence, when receiving care outside their cancer units/teams. You need to be clear. Does it mean that the non-specialist nurse works in the cancer units normally and has gone for an outreach clinic at another facility as a follow up care for adult patients with cancer? Or has the patient reported to the nearest clinic due to other health problems other than the cancer? An example could be when receiving care in other health care settings other than the oncology specialist settings. 2. Introduction Please qualify the first sentence that cancer is a commonly diagnosed non-communicable disease. 3. Replace health care workers with health care providers or professionals. 4. Page 4, lines 40-52, COVID 19 information is not relevant in this case. I suggest that it be removed. There is no proper link. 5. Page 6, Lines 20-22- should be cancer care units. communication training should be provided as part of their initial and routine training 6. Page 6, line 23, This sentence 'However, even if training is in place, evidence suggests that trained healthcare workers need help with effective communication.'....I am not sure what the author is trying to communicate.....is this the communication training that you are talking about? And when you say they need help with effective communication, would you please clarify what type of help....is it mentorship,
--

	supportive supervision?  7. Page 6, Lines 40-42.... nurses, are usually the first-line health care providers , they spend most of their time with clients and interface with cancer patients during outpatient and in-patient services 8. Page 6, Lines 45-47 Non-specialist nurses also have essential roles in cancer care which include: direct nursing care,.... 9. Review question number two to be broken into two. How is one question and when is another. 10. On PCC framework...context....will the authors include studies from any geographical location or they have a specific geographical location in mind? E.g. Africa, HIC or LIC...Please specify 11. Table 1...Exclusion criteria Studies reporting on patients that cannot communicate or have limited capacity to communicate, e.g. Dementia and Alzheimer's disease. And remove points 2,6,7 & 8 as these are only the opposites of the inclusion criteria. It is repetition.  12. Aim of the review, review questions and data to be extracted should talk to each other. As it is, the data to be extracted will not fully answer the review questions. Please revisit these items. 13. Page 13..describe what risk of bias assessment is and why it is necessary to conduct it in your review 14. Include a sample data extraction form.
--	---

VERSION 1 – AUTHOR RESPONSE

Reviewer 1 Comments

R1 Comment 1

It is advisable for the authors to incorporate the Preferred Reporting Items for Systematic Reviews and Meta-Analyses (PRISMA) flow diagram within the protocol to transparently convey the search outcomes and document the rationale behind article exclusion. Furthermore, it could be stated that an updated flow diagram will be provided upon the scoping review's completion.

Response: Thank you for pointing this out. We have included a sample PRISMA flow diagram (Please see figure 1)

“The searching, screening and selection will be documented using a PRISMA-ScR flow diagram (figure 1). The flow diagram will be updated once the review has been completed.” Page 13 line 18-22.

R1 Comment 2

I recommend that the authors consult the methodological guidance provided by Arksey and O'Malley, which delineates a six-stage framework encompassing: (1) identifying the research question; (2) identifying relevant studies; (3) selecting studies; (4) charting the data; (5) collating, summarising and reporting results; and (6) consulting with stakeholders. Additionally, referencing Levac et al.'s recommendations could serve to refine and enhance the application of this framework.

Response: We agree with the reviewer. Since the original guidance was provided by Arksey and O'Malley in 2006, it was advanced by Levac et al., and further refined by the JBI and the JBI collaboration group beginning 2015. In this protocol, we are using the JBI working group guidance.

We have reflected this in the manuscript.

“This review will be guided by the JBI methodology for scoping reviews. The JBI methodology for scoping reviews was advanced and refined from two previous scoping review guidance from Arksey and O'Malley and Levac et al..” Page 8 line 2-4

R2 Comment 3

(Page 13, line 44) The presentation of results could be expanded to include critical specifics. For instance, the methodologies for data analysis should be elaborated upon, including the tools intended for content analysis, particularly as the review includes both qualitative and quantitative research necessitating a robust mixed-methods analysis.

Response: Thank you for pointing this out. We have provided further details of the analysis. We have also provided a sample data collection tool (see table 3).

“We plan to map the evidence based on PCC using both quantitative and qualitative study designs. For quantitative designs, a narrative summary will be written to provide an overview of how the results relate to the research aims and questions. For qualitative designs, descriptive content analysis will be used” Page 15, lines 13-16

R2 Comment 4

Engaging stakeholders in consultation may yield further profound insights into the literature review. While this step is not mandatory, disseminating the findings amongst stakeholders could substantially aid in the interpretation and practical application of the results.

Response: We agree with the reviewer and we believe the methodology and findings from this scoping review may benefit from expert consultation. We have added this information in the manuscript.

“During the comprehensive literature search and data extraction we will seek additional support from subject specific experts. We will seek an expert information specialist to validate the final search strategy. We will also seek support from a cancer expert to validate the findings prior to final dissemination.” Page 16 line 3-6.

Reviewer 2

R2 Comment 1

Lines 6-15 in abstract, the part of the sentence, when receiving care outside their cancer units/teams. You need to be clear. Does it mean that the non-specialist nurse works in the cancer units normally and has gone for an outreach clinic at another facility as a follow up care for adult patients with cancer? Or has the patient reported to the nearest clinic due to other health problems other than the cancer? An example could be when receiving care in other health care settings other than the oncology specialist settings.

Response: Thank you for pointing this out. We have clarified the sentence.

“Little is known regarding how non-specialist nurses communicate with patients living with cancer when the patients are receiving care outside of their cancer units/teams.” Page 2 line 2-4

R2 Comment 2

Please qualify the first sentence that cancer is a commonly diagnosed non-communicable disease.

Response: Thank you for pointing this out. We have made this change.

“Cancer is one of the most commonly diagnosed non-communicable diseases and the leading cause of death worldwide.” Page 4 line 2-3

R2 Comment 3

Replace healthcare workers with healthcare providers or professionals.

Response: Thank you for pointing this out. We have made this change throughout the manuscript. We have used healthcare providers rather than healthcare workers.

R2 Comment 4

Page 4, lines 40-52, COVID 19 information is not relevant in this case. I suggest that it be removed. There is no proper link.

Response: Thank you for pointing this out. We have removed this paragraph.

R2 comment 5

Page 6, Lines 20-22- should be cancer care units. communication training should be provided as part

of their initial and routine training

Response: Thank you for pointing this out. We have revised this sentence as suggested.

“As such, for oncologists and other healthcare providers working in cancer units, rigorous communication training should be provided as part of their initial and routine training.” Page 6 line 3-5

R2 comment 6

Page 6, line 23, This sentence ‘However, even if training is in place, evidence suggests that trained healthcare workers need help with effective communication.’....I am not sure what the author is trying to communicate....is this the communication training that you are talking about? And when you say they need help with effective communication, would you please clarify what type of help....is it mentorship, supportive supervision?

Response: Thank you for pointing this out. We have clarified the sentence to show the kind of support needed.

“However, even if communication training is in place, evidence suggests that trained healthcare providers need help with effective communication. For example, a study in cancer care settings in Africa found that only 40% of nurses and 20% of physicians had formal communications training. Similar studies in Kenya and Belgium have also shown that lack of communication training is a major challenge in cancer care. In these studies, emphasis was placed on expanding and improving communication training during pre-service training, providing further guidance and mentorship during cancer care coordination and adapting communication to different contexts and/or cultural backgrounds.” Page 6, line 5-13

R2 Comment 7

Page 6, Lines 40-42.... nurses, are usually the first-line health care providers workers, they spend the most of their time with clients and interface with cancer patients during outpatient and in-patient services

Response: Thank you for pointing this out. We have revised this sentence as suggested.

“Nurses, including non-specialised nurses, are usually the first-line workers, they spend most of their time with clients and interface with cancer patients during outpatient and in-patient services” Page 6, line 15-17

R2 Comment 8

Page 6, Lines 45-47 Non-specialist nurses also have essential roles in cancer care which include: direct nursing care,....

Response: Thank you for pointing this out. We have revised this sentence as suggested.

“Non-specialist nurses also have essential roles in cancer care which include: direct nursing care, patient navigation, educating other healthcare providers, and care coordination” Page 6, line 18-20

R2 Comment 9

Review question number two to be broken into two. How is one question and when is another.

Response: Thank you for pointing this out. We have broken the review question into two, as suggested (page 7, line 10-20)

R2 Comment 10

On PCC framework...context....will the authors include studies from any geographical location or they have a specific geographical location in mind? E.g. Africa, HIC or LIC...Please specify
Response: Thank you for pointing this out. We will include studies from all locations of the world in the study. We have specified this in the manuscript.

“In context, we will include studies conducted in any geographical location describing all healthcare settings outside patients’ specialised cancer units or oncology teams.” Page 8, line 14-16

R2 Comment 11

Table 1...Exclusion criteria

Studies reporting on patients that cannot be able to communicate or have limited capacity to communicate, e.g. Dementia and Alzheimer's disease.And remove points 2,6,7 & 8 as these are only the opposites of the inclusion criteria. It is repetition.

Response: Thank you for pointing this out. we have removed points 2,6,7 & 8 (see Table 1, Page 9 and 10)

R2 Comment 12

Aim of the review, review questions and data to be extracted should talk to each other. As it is, the data to be extracted will not fully answer the review questions. Please revisit these items.

Response: Thank you for pointing this out. We added more information on the data that will be collected (See page 14 line 5-15). Additionally, we will adjust the data collection tool during data collection as scoping reviews are iterative in nature (see table 3).

R2 Comment 13

Page 13. Describe what risk of bias assessment is and why it is necessary to conduct it in your review

Response: Thank you for pointing this out. We have defined the risk of bias assessment and justified why we will use the assessments in this scoping review

“Risk of bias assessment is an appraisal of the methodological quality of the included studies in-order to inform the synthesis of the collected data. In this review, we will use risk of bias assessment to identify studies that should be included with the purpose of including studies with low risk of biases” page 15, line 4-8

R2 Comment 14

Include a sample data extraction.

Response: We have added table 1(sample collection tool) Page 15

R2 Comment 15

Some references are older than 10 years. where possible get current references. the old ones can only be maintained if they have some relevant information which cannot be accessed in any recently published articles.

Response: Thank you for pointing this out. We have updated the references, with references older than 2014 providing relevant methodological guidance not available in recent years.

R2 Comment 16

Include funding details.

Response: Thank you for pointing this out. We have included funding details in the funding section of the manuscript (page 21, line 19-20)

R2 Comment 17

Consider registering this protocol on Open Science Framework.

Response: Thank you for pointing this out. We have registered the protocol in Open science framework.

“We registered the protocol in Open Science Framework registry (<https://osf.io/3pmt9>)” Page 8 line 5-6

VERSION 2 – REVIEW

REVIEWER	Xu, Zhijie Zhejiang University School of Medicine Second Affiliated Hospital, Department of General Practice
REVIEW RETURNED	19-Feb-2024

GENERAL COMMENTS	The revised manuscript has addressed all my concerns.
---

REVIEWER	Majamanda, Maureen Kamuzu University of Health Sciences, Child Health Nursing
REVIEW RETURNED	29-Feb-2024

GENERAL COMMENTS	1. Page 4, lines 49 to 56, the two sentences should be combined
---

	and shortened to bring out the benefits of effective communication clearly. 2. Page 6, Lines 45-47 Non-specialist nurses also have essential roles in cancer care which include: please remove s from include. 3. Page 9 table 1, exclusion criteria point number 4 should read Studies reporting on patients that cannot communicate or have limited capacity to communicate, e.g. Dementia and Alzheimer's disease. 4. Title of tables should come first before the table. 5. PRISMA-P checklist- Please include the registration number under registration. You have included this information in the methods and analysis section. And in your responses include page and line number for easy follow up.
--	--

VERSION 2 – AUTHOR RESPONSE

Reviewer 1 Comments

R1 Comment 1

The revised manuscript has addressed all my concerns.

Response: Thank you very much.

Reviewer 2 Comments

R2 Comment 1

Page 4, lines 49 to 56, the two sentences should be combined and shortened to bring out the benefits of effective communication clearly.

Response: Thank you for pointing this out. We have merged the two sentences.

“Effective communication also helps patients with a cancer diagnosis make informed care decisions, fosters strong relationships between patients and their providers, improves clinical outcomes and improves patients' care experience.” Page 4 line 20-23

R2 Comment 2

Page 6, Lines 45-47 Non-specialist nurses also have essential roles in cancer care which include: please remove s from include.

Response: Thank you very much. We have made the change suggested by the reviewer.

“Non-specialist nurses also have essential roles in cancer care which include: direct nursing care, patient navigation, educating other healthcare providers, and care coordination” Page 6 line 18

R2 Comment 3

Page 9 table 1, exclusion criteria point number 4 should read Studies reporting on patients that cannot communicate or have limited capacity to communicate, e.g. Dementia and Alzheimer's disease.

Response: we have made the change as suggested by the reviewer.

“Studies reporting on patients that cannot communicate or have limited capacity to communicate, e.g. Dementia and Alzheimer's disease.” see table 1.

R2 Comment 4

Title of tables should come first before the table.

Response: Thank you very much for pointing this out. We have made all the changes as suggested. (please see Tables 1, 2, and 3)

R2 Comment 5

PRISMA-P checklist- Please include the registration number under registration. You have included this information in the methods and analysis section. And in your responses include page and line number for easy follow up.

Response: Thank you for pointing this out. We have added registration information in the abstract. We have also updated the checklist by including the information on registration and added page numbers (see the updated PRISMA checklist).